# Characterization of Clinical Features of Hospitalized Patients Due to the SARS-CoV-2 Infection in the Absence of Comorbidities Regarding the Sex: An Epidemiological Study of the First Year of the Pandemic in Brazil

**DOI:** 10.3390/ijerph19158895

**Published:** 2022-07-22

**Authors:** Nathália Mariana Santos Sansone, Letícia Rogini Pereira, Matheus Negri Boschiero, Felipe Eduardo Valencise, Andréa Melo Alexandre Fraga, Fernando Augusto Lima Marson

**Affiliations:** 1Laboratory of Cell and Molecular Tumor Biology and Bioactive Compounds, University of São Francisco, Bragança Paulista, São Paulo 12916-9003, Brazil; nathaliasansone@hotmail.com (N.M.S.S.); leticiarogini@gmail.com (L.R.P.); boschiero.matheus@gmail.com (M.N.B.); felipe.valencise@gmail.com (F.E.V.); 2Laboratory of Human and Medical Genetics, University of São Francisco, Bragança Paulista, São Paulo 12916-9003, Brazil; 3Department of Pediatrics, University of Campinas, Campinas, São Paulo 13083-887, Brazil; afraga@unicamp.br

**Keywords:** androgens, Brazil, COVID-19, epidemiology, SARS-CoV-2, sex

## Abstract

The male sex, due to the presence of genetic, immunological, hormonal, social, and environmental factors, is associated with higher severity and death in Coronavirus Disease (COVID)-19. We conducted an epidemiological study to characterize the COVID-19 clinical profile, severity, and outcome according to sex in patients with the severe acute respiratory syndrome (SARS) due to the fact of this disease. We carried out an epidemiological analysis using epidemiological data made available by the OpenDataSUS, which stores information about SARS in Brazil. We recorded the features of the patients admitted to the hospital for SARS treatment due to the presence of COVID-19 (in the absence of comorbidities) and associated these characteristics with sex and risk of death. The study comprised 336,463 patients, 213,151 of whom were men. Male patients presented a higher number of clinical signs, for example, fever (OR = 1.424; 95%CI = 1.399–1.448), peripheral arterial oxygen saturation (SpO_2_) < 95% (OR = 1.253; 95%CI = 1.232–1.274), and dyspnea (OR = 1.146; 95%CI = 1.125–1.166) as well as greater need for admission in intensive care unit (ICU, OR = 1.189; 95%CI = 1.168–1.210), and the use of invasive ventilatory support (OR = 1.306; 95%CI = 1.273–1.339) and noninvasive ventilatory support (OR = 1.238; 95%CI = 1.216–1.260) when compared with female patients. Curiously, the male sex was associated only with a small increase in the risk of death when compared with the female sex (OR = 1.041; 95%CI = 1.023–1.060). We did a secondary analysis to identify the main predictors of death. In that sense, the multivariate analysis enabled the prediction of the risk of death, and the male sex was one of the predictors (OR = 1.101; 95%CI = 1.011–1.199); however, with a small effect size. In addition, other factors also contributed to this prediction and presented a great effect size, they are listed below: older age (61–72 years old (OR = 15.778; 95%CI = 1.865–133.492), 73–85 years old (OR = 31.978; 95%CI = 3.779–270.600), and +85 years old (OR = 68.385; 95%CI = 8.164–589.705)); race (Black (OR = 1.247; 95%CI = 1.016–1.531), *Pardos* (multiracial background; OR = 1.585; 95%CI = 1.450–1.732), and Indigenous (OR = 3.186; 95%CI = 1.927–5.266)); clinical signs (for instance, dyspnea (OR = 1.231; 95%CI = 1.110–1.365) and SpO_2_ < 95% (OR = 1.367; 95%CI = 1.238–1.508)); need for admission in the ICU (OR = 3.069; 95%CI = 2.789–3.377); and for ventilatory support (invasive (OR = 10.174; 95%CI = 8.803–11.759) and noninvasive (OR = 1.609; 95%CI = 1.438–1.800)). In conclusion, in Brazil, male patients tend to present the phenotype of higher severity in COVID-19, however, with a small effect on the risk of death.

## 1. Introduction

During the Coronavirus Disease (COVID)-19 pandemic, the literature describes differences in the severity of the disease and mortality rates between male and female patients. According to Gebhard et al. (2020), the male sex was associated with higher hospital admissions and ~60% deaths due to the fact of COVID-19 [1]. Li et al. 2021 in a meta-analysis including 212 studies from 11 countries/regions and 281,461 individuals, demonstrated that 51.8% of the patients were men and that these patients were more frequently found among those who were severely affected by the disease (60.8%). However, the authors did not find differences in the proportion of men concerning death [2]. A study in New York with 5700 patients demonstrated that 60.3% of the individuals admitted to hospitals were men. The mortality rate was higher in this group compared to the group of female patients. The percentages of death varied according to the age group, ranging between 4.6% (30 to 39 years old) and 63.6% (+90 years old) in men and from 1.8% (20 to 29 years old) to 48.1% (80 to 89 years old) in women [3]. This finding was similar to that observed in Wuhan, China, where 58.1% of the cases affected male patients and 57.8% were more severely affected by COVID-19 [4].

The innate immune can respond to the difference in phenotype among patients with COVID-19 due to the fact of interferons’ (IFNs) action, which limits the viral infection, starts the tissue repair, and programs the adaptive immune system to eliminate the pathogen [5]. Adult women develop a fast and aggressive innate and adaptive immune response to combat the invading pathogens, while men show an attenuated immune response and are more susceptible to viral infections, since elements of the response to androgens and estrogens reside in several genes of the innate immunological system [5]. A study carried out by Takahashi et al. (2020) reported a more significant increase in interleukin (IL)-8 and C-C motif chemokine ligand 5 (CCL5) during COVID-19 development in men when compared to women. The same study evidenced that those female COVID-19 patients presented a higher number of activated T cells, differentiated from those in male patients, mainly CD8+ T cells [6]. Moreover, the Toll-like receptor (TLR)-7 is located in dendritic cells and responds to viral RNA. The gene responsible for the TLR-7 expression is located in the X chromosome and has higher expression in women. The hormonal factor cannot be disregarded, since the androgens are immunosuppressors and reduce the presence of the Toll-like receptor (TLR)-4 in macrophages, reducing the propensity to activate the innate immune response [7]. In addition, angiotensin-converting enzyme 2 (ACE2), which is necessary for SARS-CoV-2 to enter the cells, had its activity increased by male sexual hormones in an animal model [5,8].

In addition to genetic, immunological, and hormonal factors, social aspects might result in higher risk and vulnerability in male patients. The first is that the male sex identity is usually socially built to occupy public spaces so that individuals perform their work activities in conditions that make them more susceptible to the SARS-CoV-2 infection [9]. Another important factor is behavioral, since men are more likely to become involved in riskier behavior compared to women, who usually show more outstanding care for their health due to the more significant concern with their socioeconomic well-being and family responsibilities [10,11]. Taking care of their health is not common among men, who tend to seek medical assistance only after reaching a more severe condition [9]. It seems relevant to emphasize the different cultural and social behavior observed among men and women and the adhesion to hygiene practices, such as washing hands, which are carried out more carelessly or less often by men [12,13].

Some studies have considered the male sex as a risk factor for the severity of COVID-19. However, none of them compared the two groups (i.e., male and female) after excluding morbidities and other confounding factors for higher severity concerning the SARS-CoV-2 infection in hospitalized patients due to the presence of severe acute respiratory syndrome (SARS). For this reason, we aimed to associate patients’ features (i.e., demographic data, clinical signs, evolution during hospital treatment, and outcome) with COVID-19 according to sex in patients without known morbidities.

## 2. Materials and Methods

We conducted an epidemiological analysis using epidemiological data available on the OpenDataSUS (https://opendatasus.saude.gov.br/, accessed on 8 April 2021). We recorded the patients’ features using data supplied by the Brazilian Health Ministry found in the Síndrome Respiratória Aguda Grave—SRAG (Severe Acute Respiratory Syndrome, SARS) surveillance and Informação de Vigilância Epidemiológica da Gripe-SIVEP-Flu (Influenza Epidemiological Surveillance Information) systems. Patients admitted to hospital with SARS were analyzed according to their characterization for the following variables: sex, age, race, schooling, place of residence, residence in a place where an outbreak of Influenza syndrome occurred, presence of nosocomial infection, clinical symptoms, use of antiviral drugs to treat flu symptoms, need for intensive care unit (ICU) treatment and ventilatory support, thorax X-ray and high-resolution computed tomography of the lungs results, closing criteria, and outcome. Race was classified using the Brazilian Institute of Geography and Statistics definition, which groups Brazilian citizens into five official races: White, Black, individuals with multiracial backgrounds (*Pardos*), Asian, and Indigenous peoples. The race was self-declared, and individuals should identify themselves by selecting only one category.

In order to tease apart the unique effect sex has on the outcomes of COVID-19, we excluded patients with comorbidities such as cardiopathy, hematologic disorder, Down syndrome, hepatic disorder, asthma, diabetes mellitus, neurological disorder, systemic arterial hypertension, chronic respiratory disorders, immunosuppressive disorder, renal disease, obesity, and others (excluding the previous ones). The comorbidities were attributed in the dataset by a health professional.

We carried out the statistical analysis employing the Chi-square test to verify the associations between the markers evaluated and sex in hospitalized patients due to the fact of COVID-19. We included the odds ratio (OR) calculation analysis and a 95% confidence interval (95%CI). After performing the bivariate analysis, we carried out the multivariate analysis considering death or hospital discharge as an outcome. Other markers that presented significant value in the bivariate analysis for sex were considered predictor factors in the multivariate analysis. We chose the backward stepwise method in the logistic regression model. The alpha error value adopted was 0.05 in all statistical analyses carried out. We reported the study data by absolute and relative frequencies.

We carried out the Chi-square test and the multivariate analysis using the software IBM Statistical Package for the Social Sciences (SPSS) Statistics for Macintosh, Version 27.0. We employed the software OpenEpi: Open-Source Epidemiologic Statistics for Public Health to perform the OR calculations and the 95%CI. We built the figures using the software GraphPad Prism version 8.0 for Mac, GraphPad Software, La Jolla, California, USA, www.graphpad.com, accessed on 1 June 2022.

The data used in this study were publicly available. Therefore, the study was exempt from consent for not presenting risks to the research participants.

## 3. Results

We present the patients’ inclusion in the study below (Figure 1). When analyzing the distribution of patients in hospital treatment for SARS due to the fact of COVID-19 in Brazil, we observed a predominance of the male sex to the female sex in all country states. In total, 336,463 patients were hospitalized, and 213,151 were men. The states that presented the highest number of male patients were São Paulo (29.2%), Rio de Janeiro (9.0%), and Minas Gerais (6.9%) (Appendix A).

In Brazil, the first COVID-19 patient reported was a man in the 8th epidemiological week. From the 11th week onwards, the reports were predominantly male patients, reaching 60.0% to 68.6% of the notifications. The week of first clinical signs also showed a predominance of the male sex from the 10th week of notification onwards, ranging from 59.2% to 66.5% (Appendix A; Figure 2).

We summarize the comparison between male and female patients in Table 1, Table 2 and Table 3; the descriptive data are also shown in Appendix A. Men corresponded for 63.4% of the patients, and the most frequent age group infected by the SARS-CoV-2 was between 25 and 60 years old (60.4%). The most affected race was the White group (47.8%), patients that had completed high school (36.6%), and those who lived in an urban area (94.7%). Most patients lived in areas where no flu syndrome outbreak occurred (69.2%) and did not present nosocomial infection (98.5%). The most common clinical symptoms were cough (81.0%), dyspnea (77.0%), fever (74.2%), respiratory distress (67.9%), and saturation drop (peripheral arterial oxygen saturation <95%; 65.9%). Curiously, 28.4% of the patients required ICU admission, 53.7% needed noninvasive ventilatory support, and 14.3% required invasive ventilatory support. Finally, 26.3% of the patients died (Appendix A).

Male patients affected by COVID-19 were more frequently described as 25–60 years old (OR = 1.398; 95%CI = 1.308–1.495) and 61–72 years old (OR = 1.124; 95%CI = 1.050–1.203) when compared to female patients. However, there were fewer male patients in the age groups 1–12 years old (OR = 0.796; 95%CI = 0.732–0.866), 13–24 years old (OR = 0.856; 95%CI = 0.792–0.924), and +85 years old (OR = 0.692; 95%CI = 0.642–0.745) when compared to patients aged below one-year-old. Regarding race, the number of male patients, when compared with female patients, was higher in the following groups: Black (OR = 1.215; 95%CI = 1.169–1.264), Asian (OR = 1.093; 95%CI = 1.018–1.172), and *Pardos* (multiracial background; OR = 1.054; 95%CI = 1.037–1.071) using as a reference the White race. Patients that had completed elementary school (2nd stage) (OR = 1.088; 95%CI = 1.046–1.132) and those that completed high school (OR = 1.055; 95%CI = 1.021–1.090) were more frequently men, while illiterate patients (OR = 0.828; 95%CI = 0.782–0.877) presented lower numbers when compared to those that completed higher education. Men outnumbered women residing where a flu syndrome outbreak occurred (OR = 1.063; 95%CI = 1.042–1.083) (Table 1; Figure 2).

The most frequent clinical signs described in male patients in hospital treatment due to the fact of COVID-19 were fever (OR = 1.424; 95%CI = 1.399–1.448), peripheral arterial oxygen saturation < 95% (OR = 1.253; 95%CI = 1.232–1.274), chills (OR = 1.208; 95%CI = 1.107–1.317), dyspnea (OR = 1.146; 95%CI = 1.125–1.166), cough (OR = 1.132; 95%CI = 1.111–1.154), inappetence (OR = 1.080; 95%CI = 1.024–1.138), respiratory distress (OR = 1.075; 95%CI = 1.057–1.093), and myalgia (OR = 1.056; 95%CI = 1.032–1.081) when compared to females (Table 2; Figure 3). Men were also more frequently associated with the use of antiviral drugs to treat flu symptoms (OR = 1.114; 95%CI = 1.087–1.142) (Table 3; Figure 3).

When thorax X-rays were analyzed, male patients presented a higher frequency of interstitial infiltrate patterns (OR = 1.178; 95%CI = 1.166–1.244), consolidations (OR = 1.347; 95%CI = 1.243–1.460), image mixed patterns (OR = 1.214; 95%CI = 1.128–1.306), and other findings (OR = 1.291; 95%CI = 1.218–1.368) than female patients. The computerized thorax tomography with a typical COVID-19 pattern (OR = 1.129; 95%CI = 1.083–1.176) was more commonly found in male patients (Table 3; Figure 3) when compared to female patients.

When admitted for hospital treatment, male patients presented a greater need for ICU admission (OR = 1.189; 95%CI = 1.168–1.210) and use of invasive (OR = 1.306; 95%CI = 1.273–1.339) and noninvasive ventilatory support (OR = 1.238; 95%CI = 1.216–1.260) than female patients. Moreover, the male sex was associated only with a small increase in the risk of death when compared with the female sex (OR = 1.041; 95%CI = 1.023–1.060) (Table 3; Figure 3).

The multivariate analysis predicted death risk (*p*-value < 0.001; Nagelkerke R^2^ = 0.406), and male sex was one of the factors associated with this risk (OR = 1.101; 95%CI = 1.011–1.199), however, with a small effect size. In addition, other factors also contributed as death predictors such as older age (61–72 years old (OR = 15.778; 95%CI = 1.865–133.492), 73–85 years old (OR = 31.978; 95%CI = 3.779–270.600), and +85 years old (OR = 68.385; 95%CI = 8.164–589.705)), race (Black (OR = 1.247; 95%CI = 1.016–1.531), *Pardos* (multiracial background; OR = 1.585; 95%CI = 1.450–1.732), and Indigenous peoples (OR = 3.186; 95%CI = 1.927–5.266)), schooling (illiterate (OR = 2.066; 95%CI = 1.682–2.538), initial years of elementary school (1st stage) (OR = 1.728; 95%CI = 1.496–1.997), final years elementary school (2nd stage) (OR = 1.497; 95%CI = 1.292–1.736), and high school (OR = 1.232; 95%CI = 1.076–1.411)), residing in an area where an Influenza syndrome outbreak occurred (OR = 1.113; 95%CI = 1.013–1.222), clinical signs (dyspnea (OR = 1.231; 95%CI = 1.110–1.365) and peripheral arterial oxygen saturation < 95% (OR = 1.367; 95%CI = 1.238–1.508)), need for admission in ICU (OR = 3.069; 95%CI = 2.789–3.377), and need for ventilatory support (invasive (OR = 10.174; 95%CI = 8.803–11.759) and noninvasive (OR = 1.609; 95%CI = 1.438–1.800)) (Appendix A; Figure 4).

## 4. Discussion

An overview of the main factors associated with male patients’ predisposition to present the phenotype of COVID-19 with higher severity and greater risk of death is illustrated in Figure 5.

### 4.1. General Aspects

Male sex is associated with higher severity, higher death rates, and admission to ICU as a consequence of the SARS-CoV-2 infection [1,3,4,14]. This fact was observed in our patients, even in the absence of comorbidities but with a small effect size. In the literature, these findings might be associated with differences between sexes regarding the presence of sexual dimorphism; genetic, hormonal, behavioral, and social factors; age; pregnancy; menopause; immunological response [1,15,16,17]. Interestingly, COVID-19 is not the only disease affecting men and women differently. Many reports have shown that illnesses caused by other viruses, bacteria, fungi, and even parasites can produce different outcomes between sexes, and men usually present worse outcomes [14,18,19,20,21].

Men were also more affected in previous coronavirus outbreaks caused by the Middle East respiratory syndrome coronavirus (MERS-CoV). During the outbreak, a higher infection rate was observed among men and higher mortality rates in South Korea and Saudi Arabia, even if more women had been exposed to the virus [22,23,24]. Although SARS-CoV-2 presents a more remarkable genetic similarity with SARS-CoV, it also shares many common characteristics with MERS-CoV, which places both viruses in the same family and genus [25]. Therefore, it was reasonable to expect that SARS-CoV-2 would have more severe effects on men, since this had already occurred during the MERS-CoV outbreak. Curiously, other viruses responsible for pandemics, such as the Influenza virus, produced worse outcomes in women than men [26].

### 4.2. Epidemiology

Regarding the different infection rates between sexes, a recent meta-analysis reported that men and women are affected in the same proportion, even though several protective measures, such as the use of mask and gel alcohol, hand washing, and seeking medical assistance, are more common among women [14,27,28,29]. However, men presented higher mortality rates for COVID-19 and a more frequent need for ICU than women [14]. In Italy, which was one of the countries with the highest COVID-19 infection and case fatality rates, the population infection rate was slightly higher in men (55.7%) as well as the case fatality rate (13.3%)—twice as much as that recorded for women (7.4%), with a 69.1% death prevalence among men [30]. Likewise, in the United States of America (USA), a study identified that men affected by COVID-19 were at greater risk of the need for ICU admission and invasive mechanical ventilation and mortality [31]. In Brazil, Souza et al., 2020 demonstrated a higher proportion of COVID-19 cases in patients over 50 years old; in this group, 67.1% were men, and male patients accounted for ~59% of deaths [32]. Another Brazilian study developed by Zeiser et al., 2022 observed a higher hospitalization rate and need for intensive care due to the presence of COVID-19 among male patients [33].

Among the organs affected by COVID-19, the heart, digestive tract, and kidneys with acute kidney injury (ARI) are some of the most affected in more severe patients, indicating a worse prognosis [34]. In an analysis performed by He et al., 2022, a higher incidence of ARI in men infected with SARS-CoV-2 was observed compared to females [34]. In addition, the study showed that the mortality rate from COVID-19 was related to comorbidities that could induce or increase the incidence and progression of ARI, including male sex and advanced age [34]. According to Arslani et al. (2022), female patients required less hospitalization, invasive ventilatory support, and hemodynamic support compared to men [35]. In addition, the incidence of adverse clinical outcomes in patients with confirmed SARS-CoV-2 infection, such as ICU admission, readmission for respiratory distress, and death within 30 days, was almost half in female patients when compared with males [35] in a study carried out in the Kashmir region. The number of men admitted to the hospital was nearly twice that of women (66.7% vs. 33.3%) [36].

Interestingly, although the same study reported more deaths among women, it did not reach statistical significance [36]. Another study performed in the Catalonia Region from Spain, which comprised more than 17,000 patients, observed the worst outcomes in men, such as higher 30-day mortality and also a higher need for ICU treatment [37]. Finally, a study performed in Saudi Arabia comprised nearly 600 patients in which male sex was independently associated with enhanced mortality [38]. In our study, we identified a small effect of male sex in the risk of death; this was observed in the group of patients without known comorbidities.

In general, our study is in accordance with the current global and Brazilian literature, which reports higher mortality due to the presence of COVID-19 in male patients when compared to women; however, we observed only a small effect size, maybe due to the exclusion of patients with comorbidities. In addition, globally, evidence suggests that COVID-19 results in higher mortality among men in most countries, except for India, Nepal, Vietnam, and Slovenia, where more women die than men. These countries might have recorded higher mortality among women due to the biases in sex identification or more significant risks already threatening women in these countries due to the presence of demographic factors or even events related to local health profiles [39,40,41].

Although several studies, including ours, observed worse outcomes of COVID-19 in male patients, these data should be interpreted carefully, since several other characteristics might also play a role in the COVID-19 prognosis apart from gender. For instance, a recent report observed a higher mortality rate from COVID-19 in males in USA counties [42]. However, even before the pandemic, USA men had higher all-cause mortality rates than women [42], and perhaps COVID-19 did not change this dynamic, since another report observed a similar increase in crude excess in both sexes in 2020 [43]. Several other factors might contribute to the worse COVID-19 prognosis in men, such as a higher rate of comorbidities in men, especially cardiovascular disease [44]; and the fact women, especially in the USA, are more tested than men, which increases the mild and asymptomatic number of cases, thus artificially decreasing the case fatality rate (which is calculated by the number of deaths divided by the number of cases) [42,45].

### 4.3. Genetic Aspects

Zhao et al., 2020 reported that men present higher ACE2 expression in the lungs compared to women. Both ACE2 and transmembrane serine protease 2 (TMPRSS2) were proposed as modulators of different susceptibility to SARS-CoV-2 in both sexes. ACE2 is a transmembrane protein expressed on the surface of several body cells such as the heart, endothelium, intestine, kidneys, and the respiratory system epithelium. The SARS-CoV-2 enters the cell due to the virus spike (S) protein bond with the ACE2 receptor in human tissues [46,47,48]. The TMPRSS2, in turn, is a necessary protease in the S protein cleavage on S1/S2 and S2 sites, favoring the bond and fusion of the virus to cell membranes. Thus, several types of vaccines and antiviral drugs might be S protein-based. Therefore, the differentiated expression of each component associated with the virus cycle might determine different responses to the pathogenic agent resulting in higher severity and death risk [46,49].

The *ACE2* gene presents several genetic variants that can modulate the protein’s expression with the same name. Genetic variants, in some cases, do not interfere directly with the functioning of ACE proteins, since they are located outside its catalytic site. However, these variants might alter the molecule’s three-dimensional structure, acting on the virus bond and modulating its entrance into the cells [50,51]. The *ACE2* gene is located in the X chromosome (Xp22.2). It has been established that one of the X chromosomes in female mammal cells undergoes an inactivation process during differentiation. However, some genes might escape this X inactivation; *ACE2* is one of them [52]. In addition, the second X chromosome might protect women from more deleterious variants of the gene, which makes COVID-19 more aggressive in men [50,51]. In addition, when investigating X chromosome expression, ACE2 expression is slightly greater in males than in female tissues, contributing to higher infection severity in men when compared to women [52,53].

The renin–angiotensin–aldosterone system presents two types: ACE dependent and another that is not. In the ACE-dependent type, angiotensin I is converted into angiotensin II, which bonds to its AT1R receptor (angiotensin II type 1 receptor), increasing blood pressure due to the fact of its vasoconstrictor effect. In the ACE nondependent type, angiotensin I is converted into angiotensin (1-9) and angiotensin II into angiotensin (1-7), which interacts with the MAS receptor and the AT2R receptor (angiotensin II type 2 receptor) [54,55]. When the MAS receptor is stimulated, the release of nitric oxide occurs, causing vasodilation and blood pressure reduction, resulting in a protective effect [54,55,56,57]. The ACE2 expression might be influenced by estrogen all over the body, and the ACE/ACE2 ratio might deviate from the ACE2/Ang-1-7 receptor/MAS axis in women. The expression might explain why women show less severe COVID-19. ACE2 might be pro-inflammatory and pro-oxidant, while ACE-1 might mediate anti-oxidant and anti-inflammatory effects [50,58]. Structurally, SARS-CoV-2 can bond to 16 of the 20 ACE2 residues, and this usually occurs in men; however, since some women express two ACE2 genes, the chance of these two genes being identical is low, which makes the virus bond to only one of them. The variability in *ACE* expression lets the ACE2 not linked to the virus free to convert angiotensin II into angiotensin 1-7, thus reducing the chance of pulmonary edema occurring during COVID-19 [59,60,61]. Therefore, ACE2 is likely to play an essential role in determining women’s protection against the SARS-CoV-2 infection and higher COVID-19 severity.

In addition, considering genetic aspects, McCoy et al. (2021) grouped 65 patients according to the number of CAG repeats in the androgen receptor gene related to the androgenic sensitiveness. Out of the 65 patients, 31 presented less than 22 CAG repeats, while 34 presented over 22 CAG repeats, with the observation of a hospitalization time of 25 (45.2% in ICU) and 47.5 days (70.6% in ICU), respectively, indicating that those who have more than 22 CAG repeats, might have a more severe disease [62].

### 4.4. Immune Aspects

The innate immune system activation is fundamental for protection against several viruses, including SARS-CoV-2, and it is the first to be activated when some type of viral infection is detected. Women tend to show a more aggressive innate immunological response to fight pathogens, mainly viruses, when compared to men. Male patients tend to present a poor immune response and are more susceptible to viral infections [5,7,21].

Several cytokines participate in the innate immune response activation process, especially the type 1 interferon (INF-1), mainly the INFα, which is responsible for limiting the viral infection, starting the tissue repair, and acting in the adaptive immune system programming to better fight the viral infection. Even if a robust and timely INF-1 response is considered protective, an irregular production of this cytokine might contribute to its malfunction in the SARS-CoV infection, including the appearance of complications such as SARS [5,63,64]. Likewise, men seem to express higher concentrations of interleukins (IL)-8 and IL-18 and ligand 5 of the chemokine CC (CCL-5) during the SARS-CoV-2 infection, which is associated with the presence of nonclassical monocytes [6]. Conversely, women present a more intense T-cell activation, mainly CD8+ T, than men. Curiously, high levels of immune cytokines were associated with worse COVID-19 development in women, and the unsuitable response of T lymphocytes was related to worse COVID-19 progression in men [6].

Some studies reported that men infected by SARS-CoV-2 showed higher expression of several other pro-inflammatory cytokines, such as IL-6, while women expressed more IL-10. The latter is an anti-inflammatory cytokine beneficial in the resolution of inflammation and tissue repair control in inflammatory diseases [65,66,67]. In addition, IL-10 seems to be associated with a milder COVID-19 phenotype exclusively in women, since the highest concentrations of IL-10 were described in women moderately affected by the disease but not in men with the same phenotype [67]. Another finding reported is that men severely affected by COVID-19 expressed at least 12 pro-inflammatory cytokines, such as IL-4, IL-8, chemokine 1 ligand (CXCL1), soluble CD4 ligand (sCD4L), macrophage inflammatory protein (MIP-1β), and monocyte chemoattractant protein-1 (MCP-1), which were not found in women with this phenotype [67]. Studies also reported that women seem to show a better response and antibody production after the COVID-19 vaccine, mainly with mRNA vaccines, which results in fewer post-vaccination infections [68,69,70].

### 4.5. Hormonal Aspects

T and B cells present estrogen receptors in their cytoplasm; when estradiol binds to these receptors, they activate humoral immunity and the production of antibodies to fight viral infections [57]. In addition, estradiol presents immunomodulator and anti-inflammatory potential, inhibiting cell migration and attenuating the cytokine storm progression in an inflammatory process, which is a relevant death cause in COVID-19 [71].

According to Aksoyalp and Nemutlu-Samur (2021), estrogen regulates ACE negatively and, at the same time, regulates ACE2, AT2 Receptor, and MAS enzyme positively. Therefore, higher ACE2 and Ang 1-7 production are observed in the female population than in men [72]. Estradiol also inhibits the TMPRSS2, which is needed for the virus to bind with ACE2 and increases disintegrin and metalloprotease 17 expression, which is involved in the ACE2 ectodomain cleavage, allowing higher levels of the soluble ACE2 that neutralize SARS-CoV-2 and prevent it from bonding with ACE2 [57].

A study developed by Garg et al., 2020 compared 293 women, 185 of them were in the premenopause period, and 108 were in the postmenopause period. The postmenopause mortality rate (19.4%) was higher when compared to that of the premenopause group (8.6%), highlighting estrogen as a protective factor [73]. However, the findings of that study must be evaluated with caution, since old age might have interfered with its results. On the other hand, testosterone, the male hormone, suppresses the immune system [72]. Dhindsa et al., 2021 associated the testosterone plasma concentration with severity in COVID-19 patients. Testosterone levels in severely affected men at hospital admission were lower in 64.9% of them than those of moderately affected male patients [74]. Rastrelli et al., 2021 evidenced that lower levels of testosterone are related to cytokine storm, worse prognosis, and higher death rates in COVID-19 patients admitted to ICU [75].

### 4.6. Social and Environmental Factors

Effective measures to prevent infection (e.g., wearing masks and using gel alcohol) are not very popular among men. This lower adhesion might have different reasons, such as human behavior, invulnerability self-perception, illness disbelief and denialism, individual responsibility underestimation, habits, and cultural belief systems, which result in the resistance of the male population to adopt the recommended protective measures or the creation of safe strategies to face COVID-19 [76]. It seems relevant to emphasize that wearing masks and social isolation are already recognized as highly efficient strategies to combat the spread of the disease. There is evidence that the chances of disseminating the virus are 2.6% for people who keep a physical distance of a meter or more. When the distance is lower than a meter, the risk of infection reaches 12.8%. Therefore, this attitude might promote changes. The more we know about the disease, and the intervention of health professionals and means of communication might raise greater awareness of the importance and efficacy of wearing masks to control and reduce the COVID-19 spread. The understanding of protective measures is important to prevent the SARS-CoV-2 dissemination since a higher spread might promote more severe cases.

In addition, several studies compared how measures against infection were accepted between men and women. Clements (2020) evidenced that women knew more regarding COVID-19 when compared to men. Women with comorbidities understood the greater risk of this disease than their male partners and were more likely to realize that COVID-19 seriously threatened their health and adhered to the restrictive measures [77,78,79].

According to Pflugeisen and Mou (2021), men presented lower chances of realizing the illness threat and agreeing with the public health recommendation. They were more likely to have contact with people outside their circle and less likely to wash their hands after such contact [80].

Finally, severe cases of COVID-19 might be related to preexisting comorbidities [14,81], such as arterial systemic hypertension and diabetes mellitus [82]. These conditions might have a higher prevalence in men around the world, mainly because men have lower rates of seeking medical assistance [83,84,85] and also because men might have a less healthy life with a higher prevalence of harmful habits to health such as smoking and alcohol consumption [14,81].

### 4.7. Limitations

The main limitation of this study was the use of a database filled in by different health professionals at the national level. Thus, it is susceptible to some filling-in bias. Despite the significant number of participants in the study, mainly considering that it was a population without known comorbidities, the inclusion of more participants would be beneficial to reduce the chance of possible biases due to the sample power and diversity of the data evaluated, especially regarding some of the variables that presented a low input of information in the system. We developed the study with Brazilian patients only and, therefore, it cannot portray the COVID-19 characteristic of impact on sex in other countries. Finally, in the multivariate analysis, we included only the patients with complete data in the dataset, reducing the sample power.

## 5. Conclusions

Male patients without known comorbidities tend to present higher severity and death rates in COVID-19 than female patients, with a small effect in the risk of death. Maybe, the higher death rates occur due to the higher ACE2 and TMPRSS2 expression levels in men, hormonal aspects, immune response activation, and social and environmental factors. Moreover, men are less likely to seek medical assistance and show low adherence to measures to prevent infection. Thus, combining these factors might lead to an unfavorable COVID-19 prognosis among the male population. Understanding these differences between sexes is necessary for identifying vulnerable populations and preparing health teams to develop specific and efficient therapeutical approaches.

## Figures and Tables

**Figure 1 ijerph-19-08895-f001:**
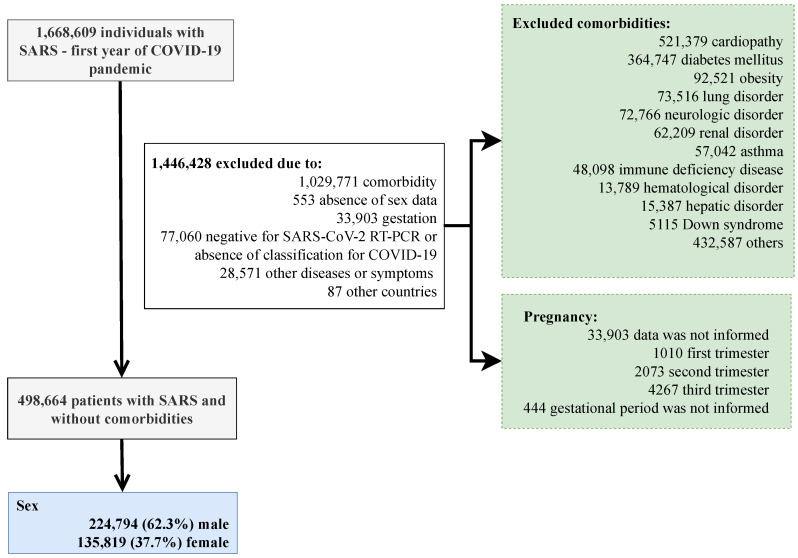
Flowchart demonstrating participants’ inclusion in the study. SARS, severe acute respiratory syndrome; COVID-19, Coronavirus Disease 2019; RT-PCR, real-time polymerase chain reaction. After exclusion, all participants had information about their sex and age groups. In order to tease apart the unique effect sex has on the outcomes of COVID-19, we excluded patients with comorbidities such as cardiopathy, hematologic disorder, Down syndrome, hepatic disorder, asthma, diabetes mellitus, neurological disorder, systemic arterial hypertension, chronic respiratory disorders, immunosuppressive disorder, renal disease, obesity, and others (excluding the previous ones).

**Figure 2 ijerph-19-08895-f002:**
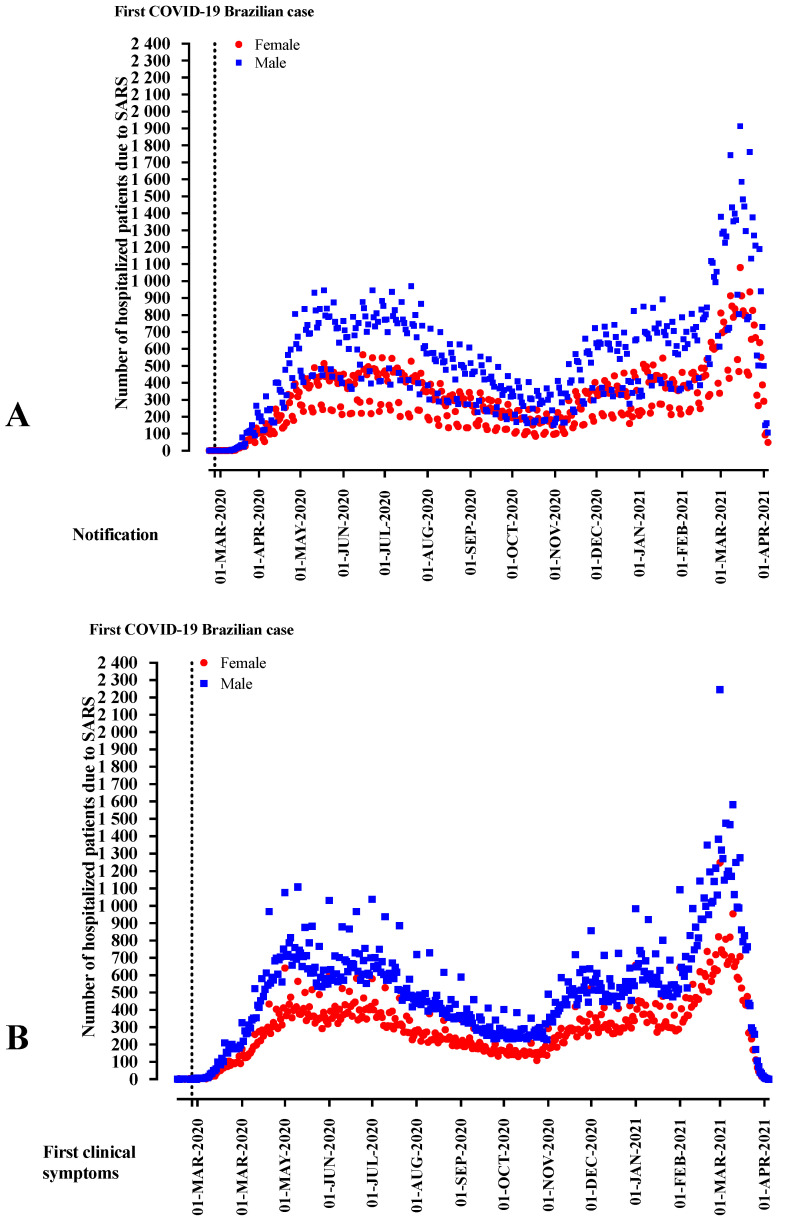
The distribution of patients hospitalized with severe acute respiratory syndrome (SARS) due to the fact of Coronavirus Disease (COVID)-19 in Brazil according to sex, weeks of disease notification, and the start of clinical signs.

**Figure 3 ijerph-19-08895-f003:**
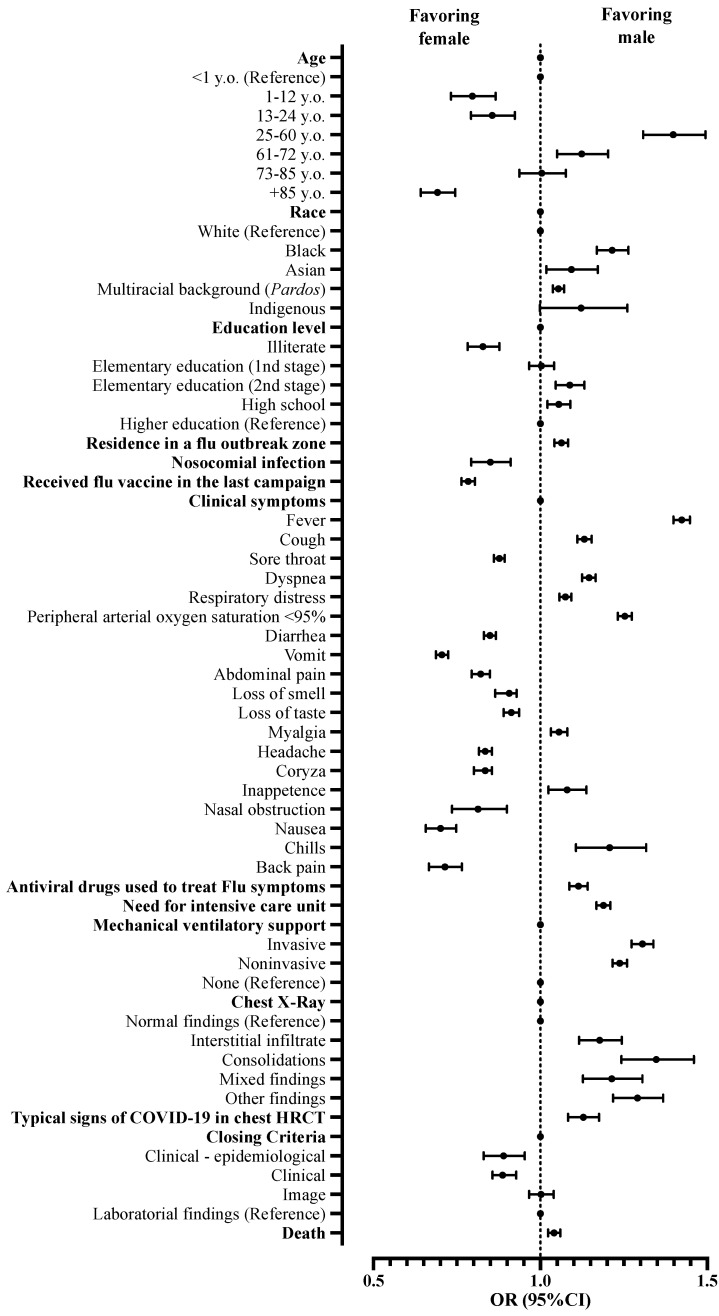
Association between characteristics of patients hospitalized with a severe acute respiratory syndrome without comorbidities in Brazil according to sex. We presented the data using odds ratios (OR) and 95% confidence intervals (95%CI). We adopted a 0.05 alpha error in all analyses. y.o., years old; HRCT, high-resolution computed tomography.

**Figure 4 ijerph-19-08895-f004:**
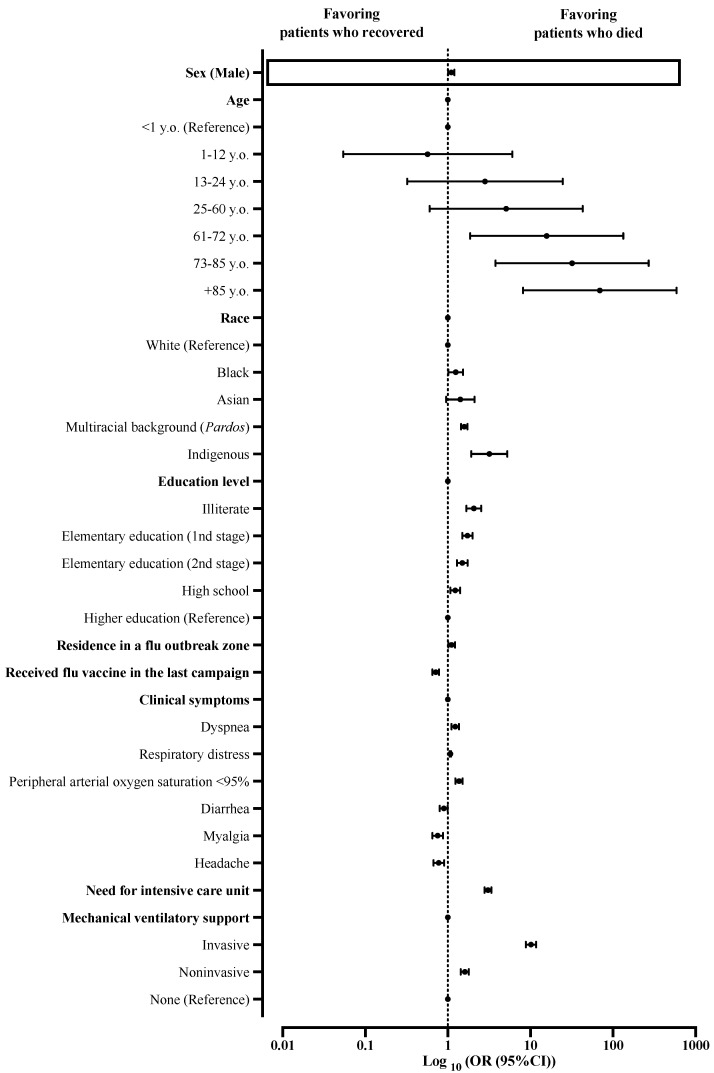
Multivariate analysis determined death predictors among patients hospitalized with severe acute respiratory syndrome without comorbidities in Brazil. We presented the data using odds ratios (ORs) and 95% confidence intervals (95%CIs). We adopted a 0.05 alpha error in all analyses. Variables inserted in step 1: sex, age, race, schooling, zone, residence in Influenza syndrome outbreak area, presence of nosocomial infection, and clinical signs (i.e., fever, cough, sore throat, dyspnea, respiratory distress, peripheral arterial oxygen saturation, diarrhea, vomit, abdominal pain, loss of smell, loss of taste, myalgia, headache, coryza, inappetence, cyanosis, nasal obstruction, vertigo, nausea, chills, sneeze, back pain, lower limb pain, flu vaccination, use of an antiviral drug to treat flu symptoms, need for intensive care treatment, and need for ventilatory support).

**Figure 5 ijerph-19-08895-f005:**
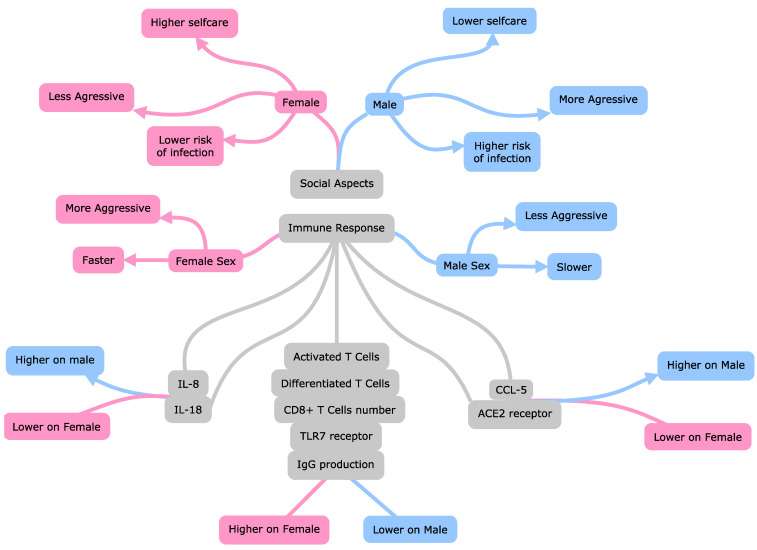
Main factors associated with higher death risk and severity in male patients with Coronavirus Disease 2019 than female patients. Interleukin 8 (IL-8); interleukin 18 (IL-18); C-C motif chemokine ligand 5 (CCL5); Toll-like receptor 7 (TLR-7); angiotensin-converting enzyme 2 (ACE2); immunoglobulin G level (IgG); cytotoxic T lymphocytes (CD8+ T).

**Table 1 ijerph-19-08895-t001:** Association between demographic markers and the presence of nosocomial infection and the sex of patients hospitalized with the severe acute respiratory syndrome (SARS) due to the fact of COVID-19 in Brazil.

Patients’ Features	Data	Male	Female	Total	*p*	OR	95%CI
Age	<1 year old	2126 (1.0%)	1499 (1.2%)	3625 (1.1%)	-	1	Reference
	1–12 years old	2990 (1.4%)	2648 (2.1%)	5638 (1.7%)	<0.001	0.796	0.732–0.866
	13–24 years old	5358 (2.5%)	4416 (3.6%)	9774 (2.9%)	<0.001	0.856	0.792–0.924
	25–60 years old	135,046 (63.4%)	68,098 (55.2%)	203,144 (60.4%)	<0.001	1.398	1.308–1.495
	61–72 years old	37,983 (17.8%)	23,820 (19.3%)	61,803 (18.4%)	<0.001	1.124	1.050–1.203
	73–85 years old	23,289 (10.9%)	16,350 (13.3%)	39,639 (11.8%)	0.917	1.004	0.937–1.076
	+85 years old	6359 (3.0%)	6481 (5.3%)	12,840 (3.8%)	<0.001	0.692	0.642–0.0745
Race	White	78,237 (47.2%)	48,634 (48.8%)	126,871 (47.8%)	-	1	Reference
	Black	8118 (4.9%)	4153 (4.2%)	12,271 (4.6%)	<0.001	1.215	1.169–1.264
	Asian	2204 (1.3%)	1254 (1.3%)	3458 (1.3%)	0.014	1.093	1.018–1.172
	*Pardos*	76,424 (46.1%)	45,081 (45.3%)	121,505 (45.8%)	<0.001	1.054	1.037–1.071
	Indigenous	794 (0.5%)	440 (0.4%)	1234 (0.5%)	0.058	1.122	0.999–1.261
Schooling	Illiterate	3437 (4.5%)	2601 (5.6%)	6038 (4.9%)	<0.001	0.828	0.782–0.877
	Elementary school (1st stage)	14,820 (19.6%)	9256 (19.8%)	24,076 (19.7%)	0.861	1.003	0.967–1.041
	Elementary school (2nd stage)	12,525 (16.5%)	7214 (15.4%)	19,739 (16.1%)	<0.001	1.088	1.046–1.132
	High school	28,139 (37.1%)	16,718 (35.8%)	44,857 (36.6%)	0.001	1.055	1.021–1.090
	Higher education	14,432 (19.0%)	9045 (19.3%)	23,477 (19.2%)	-	1	Reference
	Not applicable	2420 (3.2%)	1916 (4.1%)	4336 (3.5%)	-	-	-
Place of residence	Urban	174,266 (94.6%)	102,249 (94.8%)	276,515 (94.7%)		1	Reference
	Rural	9249 (5.0%)	5323 (4.9%)	14,572 (5.0%)	0.275	1.019	0.985–1.055
	Peri-urban	620 (0.3%)	340 (0.3%)	960 (0.3%)	0.319	1.070	0.937–1.222
Residing in an Influenza syndrome outbreak area	Yes	40,896 (31.2%)	23,296 (30.0%)	64,192 (30.8%)	<0.001	1.063	1.042–1.083
	No	89,993 (68.8%)	54,469 (70.0%)	144,462 (69.2%)		1	Reference
Presence of nosocomial infection	Yes	1994 (1.4%)	1381 (1.7%)	3375 (1.5%)	<0.001	0.850	0.793–0.911
	No	139,316 (98.6%)	82,026 (98.3%)	221,342 (98.5%)		1	Reference
Flu vaccination in the last campaign	Yes	18,422 (21.5%)	13,571 (25.9%)	31,993 (23.1%)	<0.001	0.784	0.764–0.804
	No	67,346 (78.5%)	38,869 (74.1%)	106,215 (76,9%)	-	1	Reference

OR, Odds ratio; 95%CI, 95% confidence interval. We presented the data using absolute (N) and relative (%) frequencies. We conducted the statistical analysis using the Chi-square test and adopted the 0.05 value for the alpha error.

**Table 2 ijerph-19-08895-t002:** Association between clinical signs and the sex of patients hospitalized with the severe acute respiratory syndrome (SARS) due to the fact of Coronavirus Disease (COVID)-19 in Brazil.

Clinical Signs	Data	Male	Female	Total	*p*	OR	95%CI
Fever	Yes	138,913 (76.7%)	71,334 (69.8%)	210,247 (74.2%)	<0.001	1.424	1.399–1.448
	No	42,228 (23.3%)	30,873 (30.2%)	73,101 (25.8%)		1	Reference
Cough	Yes	150,059 (81.7%)	84,429 (79.8%)	234,488 (81.0%)	<0.001	1.132	1.111–1.154
	No	33,571 (18.3%)	21,381 (20.2%)	54,952 (19.0%)		1	Reference
Sore throat	Yes	44,737 (30.2%)	28,751 (33.0%)	73,488 (31.2%)	<0.001	0.877	0.861–0.893
	No	103,629 (69.8%)	58,381 (67.0%)	162,010 (68.8%)		1	Reference
Dyspnea	Yes	140,964 (77.9%)	78,403 (75.5%)	219,367 (77.0%)	<0.001	1.146	1.125–1.166
	No	39,972 (22.1%)	25,467 (24.5%)	65,439 (23.0%)		1	Reference
Respiratory distress	Yes	114,711 (68.5%)	64,836 (66.9%)	179,547 (67.9%)	<0.001	1.075	1.057–1.093
	No	52,782 (31.5%)	32,057 (33.1%)	84,839 (32.1%)		1	Reference
Peripheral arterial oxygen saturation < 95%	Yes	114,264 (67.8%)	60,320 (62.7%)	174,584 (65.9%)	<0.001	1.253	1.232–1.274
	No	54,310 (32.2%)	35,917 (37.3%)	90,227 (34.1%)		1	Reference
Diarrhea	Yes	27,272 (19.1%)	18,246 (21.7%)	45,518 (20.1%)	<0.001	0.849	0.831–0.867
	No	115,633 (80.9%)	65,648 (78.3%)	181,281 (79.9%)		1	Reference
Vomit	Yes	14,127 (10.1%)	11,299 (13.8%)	25,426 (11.5%)	<0.001	0.705	0.687–0.724
	No	125,104 (89.9%)	70,570 (86.2%)	195,674 (88.5%)		1	Reference
Abdominal pain	Yes	9191 (9.2%)	6482 (11.0%)	15,673 (9.9%)	<0.001	0.821	0.794–0.849
	No	90,372 (90.8%)	52,337 (89.0%)	142,709 (90.1%)		1	Reference
Fatigue and asthenia	Yes	42,918 (39.4%)	25,490 (39.8%)	68,408 (39.6%)	0.105	0.983	0.964–1.003
	No	65,932 (60.6%)	38,517 (60.2%)	104,449 (60.4%)		1	Reference
Loss of smell	Yes	21,298 (20.4%)	13,662 (22.1%)	34,960 (21.1%)	<0.001	0.907	0.865–0.929
	No	82,863 (79.6%)	48,209 (77.9%)	131,072 (78.9%)		1	Reference
Loss of taste	Yes	19,893 (19.4%)	12,686 (20.8%)	32,579 (19.9%)	<0.001	0.913	0.890–0.936
	No	82,757 (80.6%)	48,166 (79.2%)	130,923 (80.1%)		1	Reference
Myalgia	Yes	23,447 (16.8%)	13,137 (16.0%)	36,584 (16.5%)	<0.001	1.056	1.032–1.081
	No	116,532 (83.2%)	68,963 (84.0%)	185,495 (83.5%)		1	Reference
Headache	Yes	21,762 (15.5%)	14,829 (18.1%)	36,591 (16.5%)	<0.001	0.835	0.816–0.855
	No	118,217 (84.5%)	67,271 (81.9%)	185,488 (83.5%)		1	Reference
Coryza	Yes	5820 (4.2%)	4056 (4.9%)	9876 (4.4%)	<0.001	0.835	0.801–0.870
	No	134,159 (95.8%)	78,044 (95.1%)	212,203 (95.6%)		1	Reference
Inappetence	Yes	4012 (2.9%)	2184 (2.7%)	6196 (2.8%)	0.004	1.080	1.024–1.138
	No	135,965 (97.1%)	79,914 (97.3%)	215,879 (97.2%)		1	Reference
Nasal obstruction	Yes	897 (0.6%)	646 (0.8%)	1543 (0.7%)	<0.001	0.813	0.735–0.900
	No	139,080 (99.4%)	81,452 (99.2%)	220,532 (99.3%)		1	Reference
Vertigo	Yes	781 (0.6%)	510 (0.6%)	1291 (0.6%)	0.058	0.897	0.802–1.003
	No	139,196 (99.4%)	81,588 (99.4%)	220,784 (99.4%)		1	Reference
Prostration	Yes	2469 (1.8%)	1492 (1.8%)	3961 (1.8%)	0.358	0.970	0.909–1.035
	No	137,508 (98.2%)	80,606 (98.2%)	218,114 (98.2%)		1	Reference
Nausea	Yes	2073 (1.5%)	1724 (2.1%)	3797 (1.7%)	<0.001	0.701	0.657–0.748
	No	137,904 (98.5%)	80,374 (97.9%)	218,278 (98.3%)		1	Reference
Malaise	Yes	2081 (1.5%)	1231 (1.5%)	3312 (1.5%)	0.811	0.991	0.923–1.064
	No	137,896 (98.5%)	80,867 (98.5%)	218,763 (98.5%)		1	Reference
Chills	Yes	1568 (1.1%)	763 (0.9%)	2331 (1.0%)	<0.001	1.208	1.107–1.317
	No	138,409 (98.9%)	81,335 (99.1%)	219,744 (99.0%)		1	Reference
Chest pain	Yes	4088 (2.9%)	2429 (3.0%)	6517 (2.9%)	0.607	0.987	0.938–1.038
	No	135,889 (97.1%)	79,669 (97.0%)	215,558 (97.1%)		1	Reference
Back pain	Yes	1803 (1.3%)	1474 (1.8%)	3277 (1.5%)	<0.001	0.714	0.666–0.765
	No	138,174 (98.7%)	80,624 (98.2%)	218,798 (98.5%)		1	Reference
Arthralgia	Yes	778 (0.6%)	405 (0.5%)	1183 (0.5%)	0.051	1.127	0.999–1.272
	No	139,199 (99.4%)	81,693 (99.5%)	220,892 (99.5%)		1	Reference

OR, Odds ratio; 95%CI, 95% confidence interval. We presented the data using absolute (N) and relative (%) frequencies. We conducted the statistical analysis using the Chi-square test and adopted the 0.05 value for the alpha error.

**Table 3 ijerph-19-08895-t003:** Association between markers during hospitalization and outcome and the sex of patients hospitalized with severe acute respiratory syndrome (SARS) due to the fact of Coronavirus Disease (COVID)-19 in Brazil.

Patients’ Features	Data	Male	Female	Total	*p*	OR	95%CI
Antiviral drug to treat flu symptoms	Yes	20,998 (14.7%)	11,324 (13.4%)	32,322 (14.2%)	<0.001	1.114	1.087–1.142
	No	121,814 (85.3%)	73,200 (86.6%)	195,014 (85.8%)	-	1	Reference
Intensive care unit	Yes	50,398 (29.6%)	25,067 (26.2%)	75,465 (28.4%)	<0.001	1.189	1.168–1.210
	No	119,679 (70.4%)	70,779 (73.8%)	190,458 (71.6%)		1	Reference
Use of ventilatory support	Invasive	25,070 (14.9%)	12,850 (13.3%)	37,920 (14.3%)	<0.001	1.306	1.273–1.339
	Noninvasive	92,216 (54.9%)	49,854 (51.6%)	142,070 (53.7%)	<0.001	1.238	1.216–1.260
	Not necessary	50,571 (30.1%)	33,843 (35.1%)	84,414 (31.9%)	-	1	Reference
Thorax X-ray result	Normal	3800 (3.7%)	2520 (4.2%)	6320 (3.9%)	-	1	Reference
	Interstitial infiltrate	25,876 (25.1%)	14,563 (24.3%)	40,439 (24.8%)	<0.001	1.178	1.116–1.244
	Consolidation	2966 (2.9%)	1460 (2.4%)	4426 (2.7%)	<0.001	1.347	1.243–1.460
	Mixed	3798 (3.7%)	2075 (3.5%)	5873 (3.6%)	<0.001	1.214	1.128–1.306
	Other	14,127 (13.7%)	7257 (12.1%)	21,384 (13.1%)	<0.001	1.291	1.218–1.368
	Not carried out	52,384 (50.9%)	32,168 (53.6%)	84,552 (51.9%)	-	-	-
Thorax HRCT	Typical COVID-19	58,965 (67.8%)	31,262 (62.8%)	90,227 (65.9%)	<0.001	1.129	1.083–1.176
	Undetermined COVID-19	2110 (2.4%)	1344 (2.7%)	3454 (2.5%)	-	1	Reference
	Atypical COVID-19	1100 (1.3%)	682 (1.4%)	1782 (1.3%)	-	1	Reference
	Negative for pneumonia	231 (0.3%)	238 (0.5%)	469 (0.3%)	-	1	Reference
	Other	3341 (3.8%)	1801 (3.6%)	5142 (3.8%)	-	1	Reference
	Not carried out	21,286 (24.5%)	14,458 (29.0%)	35,744 (26.1%)	-	-	-
Closing criterion	Laboratorial	188,426 (92.6%)	108,199 (92.2%)	296,625 (92.4%)	-	1	Reference
	Epidemiological clinical	2088 (1.0%)	1348 (1.1%)	3436 (1.1%)	<0.001	0.890	0.830–0.953
	Clinical	4826 (2.4%)	3126 (2.7%)	7952 (2.5%)	<0.001	0.887	0.847–0.928
	Image clinical	8173 (4.0%)	4683 (4.0%)	12,856 (4.0%)	0.915	1.002	0.966–1.040
Outcome	Cure	130,340 (73.4%)	76,470 (74.2%)	206,810 (73.7%)	<0.001	1	Reference
	Death *	47,158 (26.6%)	26,565 (25.8%)	73,723 (26.3%)	-	1.041	1.023–1.060

* The relative risk was 1.015 (95%CI = 1.009–1.021). OR, Odds ratio; 95%CI, 95% confidence interval; HRCT, high-resolution computerized tomography. We presented the data using absolute (N) and relative (%) frequencies. We conducted the statistical analysis using the Chi-square test and adopted the 0.05 value for the alpha error.

## Data Availability

According to the SARS surveillance, the Brazilian Health Ministry input the data in the OpenDataSUS (https://opendatasus.saude.gov.br/, accessed on 8 April 2021) considering the platform of the Sistema de Informação de Vigilância Epidemiológica da Gripe-SIVEP-Gripe (Influenza Epidemiological Surveillance Information System—SIVEP-Influenza).

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
