# Peer review of "Characterization of Clinical Features of Hospitalized Patients Due to the SARS-CoV-2 Infection in the Absence of Comorbidities Regarding the Sex: An Epidemiological Study of the First Year of the Pandemic in Brazil"

_ijerph, 2022, doi:10.3390/ijerph19158895_

Round 1
Reviewer 1 Report
The manuscript is interesting but needs some major revisions
1) Introduction: the second paragraph of the Introduction (innate immune response) is too long and must be summarized
2) Methods, exclusion of patients with comorbidities: must be explained with more detail. Which comorbidities are registered in the information source, e.g. is also simple hypertension reported? Figure 1 provides some data about exclusion of study subjects, please report also how many patients were excluded by gender and age group
3) Presentation of Results: please delete Table 1. Table 2 must be converted in a Graph, showing subsequent epidemic waves on a temporal axis. Table 3: please report separate columns by sex, with the corresponding p-values. Figure 2: can be deleted, if Table 3 is modified as suggested.
4) Discussion. Epidemiology section: please remove the part about geographical variability, it is not the focus of the study. Social and environmental factors section: only factors influencing the risk of infection are discussed; instead the study is about inhospital mortality, please discuss how factors such as lower awareness in the male gender possibly influenced the severity of infection
Author Response
Comments and Suggestions for Authors
The manuscript is interesting but needs some major revisions
1) Introduction: the second paragraph of the Introduction (innate immune response) is too long and must be summarized
Reply: We Thank for the review. We changed the second paragraph of the regarding immune response, as it follows:
“The innate immune can respond to the difference in phenotype among patients with COVID-19 due to interferons (IFNs) action, which limits the viral infection, starts the tissue repair, and programs the adaptive immune system to eliminate the pathogen [5]. Adult women develop a fast and aggressive innate and adaptive immune response to combat the invading pathogens, while men show an attenuated immune response and are more susceptible to viral infections since elements of the response to androgens and estrogens reside in several genes of the innate immunological system [5]. A study carried out by Takahashi et al. (2020) reported a more significant increase in interleukins (IL)-8 and C-C Motif Chemokine Ligand 5 (CCL5) during the COVID-19 development in men when compared to women. The same study evidenced those female COVID-19 patients presented a higher number of activated T cells, differentiated from those in male patients, mainly CD8+ T cells [6]. Also, the Toll-like Receptor (TLR)-7 is located in dendritic cells and responds to viral RNA. The gene responsible for the TLR-7 expression is located in the X chromosome and has higher expression in women. The hormonal factor cannot be disregarded since the androgens are immunosuppressors and reduce the presence of the Toll-like Receptor (TLR)-4 in macrophages, reducing the propensity to activate the innate immune response. In addition, the angiotensin-converting enzyme 2 (ACE2), which is necessary for the SARS-CoV-2 to enter the cells, has its activity increased by male sexual hormones [5,7].”
2) Methods, exclusion of patients with comorbidities: must be explained with more detail. Which comorbidities are registered in the information source, e.g. is also simple hypertension reported? Figure 1 provides some data about exclusion of study subjects, please report also how many patients were excluded by gender and age group.
Reply: We thank the reviewer for the comments. We agree that we should have listed the excluded comorbidities, thus we included an excerpt in the methods section to list all the comorbidities inputted in the dataset that were excluded from our study, as follows:
“In order to tease apart the unique effect sex has in the outcomes of COVID-19, e excluded patients with comorbidities, such as cardiopathy, hematologic disorder, Down syndrome, hepatic disorder, asthma, diabetes mellitus, neurological disorder, systemic arterial hypertension, chronic respiratory disorders, immunosuppressive disorder, renal disease, obesity, and others (excluding the previous ones).”
Also, all the patients after the first exclusion criteria (see figure 1) had the information for sex and age groups. In this context, we added this information in the table footnote.
3) Presentation of Results: please delete Table 1. Table 2 must be converted in a Graph, showing subsequent epidemic waves on a temporal axis. Table 3: please report separate columns by sex, with the corresponding p-values. Figure 2: can be deleted, if Table 3 is modified as suggested.
Reply: Dear reviewer, we thank you for your important contribution. Now, we have the following tables and figures:
Table 1 was included as Supplementary Material
Table 2 was included in Figure 2
Figure 1 was not changed
Table 3 was modified as recommended and we separate it into three tables (1 to 3)
4) Discussion. Epidemiology section: please remove the part about geographical variability, it is not the focus of the study. Social and environmental factors section: only factors influencing the risk of infection are discussed; instead the study is about in hospital mortality, please discuss how factors such as lower awareness in the male gender possibly influenced the severity of infection
Reply: We thank the reviewer for the comments. We removed the part about geographical variability. We also included minor corrections in the other topics of the discussion.
Reviewer 2 Report
Brief summary The authors did an epidemiological study to characterize the COVID-19 clinical profile, severity, and outcome according to sex using the OpenDataSUS, which stores information about SARS in Brazil. The study comprised 336,463 patients, 213,151 of them were men. The inclusion of a high number of participants would be beneficial to reduce the chance of possible biases due to the sample power and diversity of the data evaluated
General concept comments
The main message is that of its title: Male sex predisposes to higher severity and is a risk factor for death in COVID-19.
The main message is based on the following statistical data that present significant odds ratios but with an effect size that many statisticians would consider an insignificant effect size (OR < 1.68):
- “The multivariate analysis enabled the prediction of risk of death, and the male sex was one of the main predictors (OR=1.101; 95%CI=1.011-1.199).”
- Men also had a greater need for admission in intensive care unit (ICU, OR=1.189; 95%CI=1.168-1.210), and the use of invasive ventilatory support (OR=1.306; 28 95%CI=1.273-1.339), and non-invasive ventilatory support (OR=1.238; 95%CI= 1216-1260).
The small intensity or strength of the association of the male sex with death can be suggested by the following data:
- a) case-fatality risk in both sexes are also very close, 26.6% in men and 25.8% in women.
- b) the relative risk of death between men and women was only 1.015 (95%CI=1.009-1.021), that is, close to 1.
There is little point in presenting effect sizes in articles if they are not interpreted and discussed. The practical significance is low since the magnitude of the effect is in a range that many scientists would consider insignificant.
However, the work is valuable and provides very interesting statistical data due to the size of its effect regarding the prediction of risk of death: older age [61-72 years old (OR=15.778; 95%CI=1.865-133.492), 73-85 years old (OR=31.978; 95%CI=3.779-270.600), and +85 years old (OR=68.385; 95%CI=8.164-589.705)]; Indigenous (OR=3.186; 95%CI=1.927-5.266)]; need for admission in the ICU (OR=3.069; 95%CI=2.789-3.377), and for ventilatory support [invasive (OR=10.174; 95%CI=8.803-11.759).
The authors carry out a detailed and interesting analysis of the genetic, immunological, hormonal, social factors that might result in higher risk and vulnerability of male patients.
In general, their study is in accordance with the current literatura which reports higher mortality due to COVID-19 in male patients when compared to women.
However, the size of the effect in the multivariate analysis to support the main messages that male sex predisposes to higher severity and is a risk factor for death in COVID-19 is very low, being at levels that some statisticians would consider irrelevant.
Therefore, the message of the title, the introduction, the discussion, as well as the general orientation of the article should change to take into account the size of the effect of the multivariate analysis.
Author Response
Comments and Suggestions for Authors
Brief summary
The authors did an epidemiological study to characterize the COVID-19 clinical profile, severity, and outcome according to sex using the OpenDataSUS, which stores information about SARS in Brazil. The study comprised 336,463 patients, 213,151 of them were men. The inclusion of a high number of participants would be beneficial to reduce the chance of possible biases due to the sample power and diversity of the data evaluated
Reply: We thank the reviewer for the comments.
General concept comments
The main message is that of its title: Male sex predisposes to higher severity and is a risk factor for death in COVID-19.
The main message is based on the following statistical data that present significant odds ratios but with an effect size that many statisticians would consider an insignificant effect size (OR < 1.68):
- “The multivariate analysis enabled the prediction of risk of death, and the male sex was one of the main predictors (OR=1.101; 95%CI=1.011-1.199).”
- Men also had a greater need for admission in intensive care unit (ICU, OR=1.189; 95%CI=1.168-1.210), and the use of invasive ventilatory support (OR=1.306; 28 95%CI=1.273-1.339), and non-invasive ventilatory support (OR=1.238; 95%CI= 1216-1260).
The small intensity or strength of the association of the male sex with death can be suggested by the following data:
- a) case-fatality risk in both sexes are also very close, 26.6% in men and 25.8% in women.
- b) the relative risk of death between men and women was only 1.015 (95%CI=1.009-1.021), that is, close to 1.
There is little point in presenting effect sizes in articles if they are not interpreted and discussed. The practical significance is low since the magnitude of the effect is in a range that many scientists would consider insignificant.
However, the work is valuable and provides very interesting statistical data due to the size of its effect regarding the prediction of risk of death: older age [61-72 years old (OR=15.778; 95%CI=1.865-133.492), 73-85 years old (OR=31.978; 95%CI=3.779-270.600), and +85 years old (OR=68.385; 95%CI=8.164-589.705)]; Indigenous (OR=3.186; 95%CI=1.927-5.266)]; need for admission in the ICU (OR=3.069; 95%CI=2.789-3.377), and for ventilatory support [invasive (OR=10.174; 95%CI=8.803-11.759).
The authors carry out a detailed and interesting analysis of the genetic, immunological, hormonal, social factors that might result in higher risk and vulnerability of male patients.
In general, their study is in accordance with the current literatura which reports higher mortality due to COVID-19 in male patients when compared to women.
However, the size of the effect in the multivariate analysis to support the main messages that male sex predisposes to higher severity and is a risk factor for death in COVID-19 is very low, being at levels that some statisticians would consider irrelevant.
Therefore, the message of the title, the introduction, the discussion, as well as the general orientation of the article should change to take into account the size of the effect of the multivariate analysis.
Reply: The authors thank the reviewer for the important contribution. In this context, we included several minor corrections in the text, mainly, regarding the small effect size of the sex on death rate in our study. For example:
Title:
“Characterization of clinical features of the hospitalized patients due to SARS-CoV-2 infection in the absence of comorbidities regarding the sex: an epidemiological study of the first year of the pandemic in Brazil”
Abstract:
“Abstract: Male sex, due to genetic, immunological, hormonal, social, and environmental factors, is associated with higher severity and death in Coronavirus Disease (COVID)-19. We did an epidemiological study to characterize the COVID-19 clinical profile, severity, and outcome according to sex in patients with the severe acute respiratory syndrome (SARS) due to this disease. We carried out an epidemiological analysis using epidemiological data made available by the OpenDataSUS, which stores information about SARS in Brazil. We recorded the features of the patients admitted to the hospital for SARS treatment due to COVID-19 (in the absence of comorbidities) and associated these characteristics with sex and risk of death. The study comprised 336,463 patients, 213,151 of whom were men. Male patients presented higher number of clinical signs, for example, fever (OR=1.424; 95%CI=1.399-1.448), arterial oxygen saturation (SpO2) <95% (OR=1.253; 95%CI=1.232-1.274), and dyspnea (OR=1.146; 95%CI=1.125-1.166); as well as greater need for admission in intensive care unit (ICU, OR=1.189; 95%CI=1.168-1.210), and the use of invasive ventilatory support (OR=1.306; 95%CI=1.273-1.339), and non-invasive ventilatory support (OR=1.238; 95%CI=1.216-1.260) when compared with female patients. Curiously, the male sex was associated only with a small increase in the risk of death when compared with the female sex (OR=1.041; 95%CI=1.023-1.060). We did a secondary analysis to identify the main predictors of death. In that sense, the multivariate analysis enabled the prediction of the risk of death, and the male sex was one of the predictors (OR=1.101; 95%CI=1.011-1.199), however, with a small effect size. Besides that, other factors also contributed to this prediction and presented a great effect size, they are listed below: older age [61-72 years old (OR=15.778; 95%CI=1.865-133.492), 73-85 years old (OR=31.978; 95%CI=3.779-270.600), and +85 years old (OR=68.385; 95%CI=8.164-589.705)]; race [Black (OR=1.247; 95%CI=1.016-1.531), Pardos (multiracial background; OR=1.585; 95%CI=1.450-1.732), and Indigenous (OR=3.186; 95%CI=1.927-5.266)]; clinical signs [for instance, dyspnea (OR=1.231; 95%CI=1.110-1.365) and SpO2 <95% (OR=1.367; 95%CI=1.238-1.508)], need for admission in the ICU (OR=3.069; 95%CI=2.789-3.377), and for ventilatory support [invasive (OR=10.174; 95%CI=8.803-11.759) and non-invasive (OR=1.609; 95%CI=1.438-1.800)]. In conclusion, in Brazil, male patients tend to present the phenotype of higher severity in COVID-19, however, with a small effect on the risk of death.”
Results
“When admitted for hospital treatment, male patients presented a greater need for ICU admission (OR=1.189; 95%CI=1.168-1.210), and use of invasive (OR=1.306; 95%CI=1.273-1.339), and non-invasive ventilatory support (OR=1.238; 95%CI=1.216-1.260) than female patients. Besides that, the male sex was associated only with a small increase in the risk of death when compared with the female sex (OR=1.041; 95%CI=1.023-1.060) (Table 3; Figure 3).”
Limitations
“The main limitation of this study is the use of a database filled in by different health professionals at the national level. Thus, it is susceptible to some filling-in bias. Despite the significant number of participants in the study, mainly considering that it was a population without known comorbidities, the inclusion of more participants would be beneficial to reduce the chance of possible biases due to the sample power and diversity of the data evaluated, especially regarding some of the variables that presented a low input of information in the system. We developed the study with Brazilian patients only, and therefore, it cannot portray the COVID-19 characteristic of impact on sex in other countries. Finally, in the multivariate analysis, we included only the patients with complete data in the dataset, reducing the sample power.”
Conclusions
“Male patients without known comorbidities tend to present higher severity and death rates in COVID-19 than female patients, with a small effect in the risk of death. Maybe, the higher death rates occur due to higher ACE2 and TMPRSS2 expression levels in men, hormonal aspects, immune response activation, and social and environmental factors. Moreover, men are less likely to seek medical assistance and show low adherence to measures to prevent infection. Thus, combining these factors might lead to an unfavorable COVID-19 prognosis among the male population. Understanding these differences between sexes is necessary for identifying vulnerable populations and preparing health teams to develop specific and efficient therapeutical approaches.”
Reviewer 3 Report
Authors have obtained an impressive resource of COVID-19 health data to study gender bias in Brazilian patient population. By the numbers they provide it appears that there is male dominance in COVID-19 disease, I, however, think that they oversimplify the results and conclusions. As shown in a recent study by Danielsen et al., Social Science and Medicine Volume 294, February 2022, 114716 gender disparity in COVID-19 is much more complex than what the investigators present. Due to the large volume of publications on gender bias in COVID-19 disease, it is difficult to be objective with the results. I am also concerned that the reference list did not include any publications from 2022. Due to the fast-moving nature of COVID-19 research, there are many new recent findings that should be carefully considered
Author Response
Comments and Suggestions for Authors
Authors have obtained an impressive resource of COVID-19 health data to study gender bias in Brazilian patient population. By the numbers they provide it appears that there is male dominance in COVID-19 disease, I, however, think that they oversimplify the results and conclusions. As shown in a recent study by Danielsen et al., Social Science and Medicine Volume 294, February 2022, 114716 gender disparity in COVID-19 is much more complex than what the investigators present. Due to the large volume of publications on gender bias in COVID-19 disease, it is difficult to be objective with the results. I am also concerned that the reference list did not include any publications from 2022. Due to the fast-moving nature of COVID-19 research, there are many new recent findings that should be carefully considered
Reply: We thank you for the review. We included references from 2022 in our study. Just like Danielsen et al. report sex bias is a combination of variations in data collection practices and sex patterns in health behaviors, occupational exposures, and pre-existing health conditions. We also added a small excerpt regarding sex bias explaining there might have several other factors apart from sex that could have contributed to worse COVID-19 prognosis in men.
Round 2
Reviewer 1 Report
The authors addressed the comments raised to the first version of the manuscript. I think that a careful copyediting is required before publication. I'm not a native English speaker, but some sentences are wrong/unclear; only as an example, see introduction, lines 73-76
Author Response
The copyediting will be done by the publisher's in-house editorial team before final publication. Thank you.
Reviewer 2 Report
The authors have followed the indicated recommendations. The content of the manuscript is interesting.
Reviewer 3 Report
I appreciate the investigators referencing the recent study I requested.